# Detection and Estimation of Retained Austenite in a High Strength Si-Bearing Bainite-Martensite-Retained Austenite Micro-Composite Steel after Quenching and Bainitic Holding (Q&B)

**Shima Pashangeh [1], Hamid Reza Karimi Zarchi [1,*], Seyyed Sadegh Ghasemi Banadkouki [1] and Mahesh C. Somani [2]**

[1]  Department of Mining and Metallurgical Engineering, Mining Technologies Research Center, Yazd University, Yazd 98195-741, Iran; pashangeh.a@gmail.com (S.P.); sghasemi@yazd.ac.ir (S.S.G.B.)

[2]  Materials and Mechanical Engineering, Centre for Advanced Steels Research, University of Oulu, 90014 Oulun Yliopisto, Finland; mahesh.somani@oulu.fi

*  Correspondence: karimizarchi@yazd.ac.ir; Tel.: +98-913-2528-680; Fax: +98-35-3821-0995

**Abstract:** To develop an advanced high strength steel with reasonable ductility based on low alloying concept as well as micro-composite microstructure essentially consisting of bainite, martensite and retained austenite, a Si-bearing, low alloy medium carbon sheet steel (DIN1.5025 grade) was subjected to typical quenching and bainitic holding (Q&B) type isothermal treatment in the bainitic region close to martensite start temperature ($M_s$) for different durations in the range 5s to 1h. While the low temperature bainite has the potential to provide the required high strength, a small fraction of finely divided austenite stabilized between the bainitic laths is expected to provide the desired elongation and improved work hardening. Various materials characterization techniques including conventional light metallography, field emission scanning electron microscopy FE-SEM, electron backscatter diffraction (EBSD), differential thermal analysis, X-ray diffraction (XRD) and vibrating sample magnetometry (VSM), were used to detect and estimate the volume fraction, size and morphology and distribution of retained austenite in the micro-composite samples. The results showed that the color light metallography technique using LePera's etching reagent could clearly reveal the retained austenite in the microstructures of the samples isothermally held for shorter than 30s, beyond which an unambiguous distinction between the retained austenite and martensite was imprecise. On the contrary, the electron microscopy using a FE-SEM was not capable of identifying clearly the retained austenite from bainite and martensite. However, the EBSD images could successfully distinguish between bainite, martensite and retained austenite microphases with good contrast. Although the volume fractions of retained austenite measured by EBSD are in accord with those obtained by XRD and color light metallography, the XRD measurements showed somewhat higher fractions owing to its ability to acquisition and analyze the diffracted X-rays from very finely divided retained austenite, too. The differential thermal analysis and vibrating sample magnetometry techniques also confirmed the stabilization of retained austenite finely divided in bainite/martensite micro-composite microstructures. In addition, the peak temperatures and intensities corresponding to the decomposition of retained austenite were correlated with the related volume fractions and carbon contents measured by the XRD analysis.

**Keywords:** low alloy Si-bearing steel; quenching and bainitic holding; micro-composite bainite-martensite-retained austenite steel

## 1. Introduction

Depending on the steel type, its chemical composition, and processing parameters affecting the heat treatment cycles, including the reheating and austenitizing conditions, subsequent cooling paths and final quenching rates and media, heat treatable low alloy steels can be designed and developed with micro-composite microstructures comprising a mixture of bainite, martensite and retained austenite with a desired combination of engineering properties in the final products [1–9]. The role of retained austenite in these micro-composite microstructures is very complex and could have both positive as well as adverse effects on the final engineering properties, depending on its size, morphology, distribution and, of course, the volume fraction. Thus, the detection and estimation of retained austenite in respect of its amount, size and distribution, besides thermal stability, have been considered to be a key factor in the design and manufacturing of these high strength micro-composite steels [7,10–14]. So far, various qualitative and quantitative materials characterization techniques have been used by several investigators to detect and measure the amount of retained austenite in the heat treatable micro-composite steels and these include conventional light optical microscopy [15–18], color metallography [14,16,18,19], X-ray diffraction (XRD) [20–27], dilatometry method [20,28–31], magnetic properties measurement [12,27,32–34], differential thermal analysis (DTA) [35,36], differential scanning calorimetry (DSC) [20,37,38], electron backscatter diffraction (EBSD) [9,14,39–44] and thin film Mössbauer spectroscopy [13,22,45,46]. In addition, atomic probe tomography (APT) [47] is another material analysis technique offering extensive capabilities for both 3D imaging and chemical composition measurements of retained austenite at the atomic scale (around 0.1–0.3 nm resolution in depth and 0.3–0.5 nm laterally) rather than just phase identification or at least estimation of its fraction. Nevertheless, it can be used for analyzing carbon and other alloying elements in retained austenite. It is also possible to detect clustering of carbon atoms or formation of carbides that may manifest during partitioning.

Conventional black-and-white [15–18] and color light metallography [14,16,18,19] techniques have generally been used by numerous researchers in order to reveal retained austenite from other microconstituents in multiphase low alloy steel microstructures. In general, ease of sample preparation, lack of limitation on exterior geometry of samples and capability in respect of determination of retained austenite volume fraction by point counting method are among the key advantages of light metallography techniques. Nevertheless, based on the steel chemical composition and the type of applied heat treatment cycles, detection of various microphases in micro-composite microstructure is quite complex, such as finely divided retained austenite from martensite, besides the limitation in respect of resolution. Hence, serious difficulties have been encountered in various cases during the preparation of metallography samples [10,14,15,18,19]. The usefulness of light metallographic detection of retained austenite in the carburized EX24, EX32 and SAE4820 low carbon low alloy steels has been reported with carbon contents in excess of about 0.6 wt% in case-hardened surface layers [10]. In addition, the traditional manual point counting method for quantification of retained austenite based on the conventional light metallography techniques is time-consuming, less accurate and tedious as well [10,14,48]. Owing to the large number of low alloy steel microstructures containing ferrite, bainite, martensite and retained austenite, identification of the microconstituents by electron microscopy techniques are often problematic. As stated in [49] one of the shortfalls of EBSD in the quantification of low alloy steel, concerns the presence of different morphologies of ferrite that may have similar crystallographic structure and hence, can lead to erroneous. Bainitic microstructures also have lower confidence indexes owing to the presence of large number of dislocations, which is responsible for the high strength and low ductility typical of this phase. To some extent, the scanning electron microscopy is able to distinguish between various morphologies of ferrite and bainite in the microstructures (see Section 3.1.2. FE-SEM micrographs). However, SEM images are often not able to identify and distinguish between martensite and adjacent retained austenite, especially in M/RA islands, as mentioned in this manuscript. However, these can be used to make a distinction between martensite matrices and bainite crystals in the low alloy microcomposite microstructures.

Generally, XRD has been widely used as a quantitative technique to measure retained austenite in micro-composite low alloy heat treatable steels [12,50–52]. The XRD analysis needs carefully prepared smooth sample surfaces without any mechanical deformation and is capable of detecting retained austenite down to nearly 0.5–1% volume fraction in the steel samples [48,53]. The main advantages of XRD method are its accuracy and ability to detect and estimate retained austenite that is finely divided in the matrix and not detectable by light optical microscopy. However, this method should be applied to low alloy micro-composite samples with nearly random crystallographic orientations of ferrite, bainite, martensite, retained austenite and carbide microphases, because any preferred crystallographic orientations of these microphases can drastically influence the XRD measurements [53]. In other cases, the micro-composite low alloy steel samples have a distinct crystallographic texture and a special step is required to compensate for this texture effects, so the method becomes somewhat complicated and time-consuming as well [10,48,53].

Thermal dilatometry method has been employed to determine the amount of retained austenite in micro-composite low alloy steels by heating a steel sample at a predetermined heating rate to a specified temperature and for a desired soaking time followed by subsequent cooling along proposed cooling paths at different rates [20,28–31]. Dilatometry measurements make it possible to measure the volume fraction of transformed retained austenite in heating through the dilatation change, but the measurement accuracy is not high enough since the knowledge of lattice parameters is limited and also the transformational plasticity may affect the net dilatation results, too. Further review of relevant literature data indicates that the vibrating sample magnetometry (VSM) [28,33,37] and differential scanning calorimetry [20,37,38] have also been used to detect retained austenite from other phases in low alloy heat treatable micro-composite steels. A VSM is a device that can measure the magnetic properties of a sample by its controlled movement inside a uniform applied magnetic field. The voltage induced by this movement can be detected by the pickup coils and is directly proportional to the extent of magnetization of the sample. The saturation magnetization Ms refers to the maximum magnetization in a material, which occurs when a large enough field is applied. The material only has one domain remaining in which all magnetic moments are aligned parallel to the magnetic field. Any difference in saturation magnetization between a sample containing some retained austenite and another without any retained austenite is directly related to the volume fraction of retained austenite phase. Therefore, the samples bearing some retained austenite will have higher saturation magnetization values in comparison with those without any retained austenite. In magnetization method, the saturation magnetization can be obtained from the maximum value of magnetization (Y-axis) in Hysteresis Loops, and hence, the difference in saturation magnetization value from that estimated on a fully ferromagnetic sample is directly related to the volume fraction of non-ferromagnetic retained austenite. This is due to the fact that ferrite, carbide, pearlite, bainite and martensite are ferromagnetic at low temperatures, while only the retained austenite is paramagnetic in the micro-composite microstructures [12,32–34]. Moreover, the coercivity, also called the magnetic coercivity (Hs), a measure of the ability of a ferromagnetic material to withstand an external magnetic field without becoming demagnetized is another magnetic parameter which can be used in order to detect the paramagnetic retained austenite in the samples microstructures. Experimentally, the coercivity can be measured by a horizontal intercept of the Hysteresis Loop [28,33,37].

The DSC method was used to study the thermal stability of retained austenite in different CMnSi and CMnAl steels under quenching and partitioning (Q&P) heat-treated condition and it has been shown that the retained austenite decomposition temperature was about 370 °C [37]. Another research work reported that the thermal stability of retained austenite in a low carbon low alloy steel can be increased to about 500 °C [35]. These arguments are still ongoing and sometimes conflicting results have been reported concerning the volume fraction of retained austenite stabilized in low alloy heat-treated micro-composite steels [12,14,15,32,54]. Notwithstanding these arguments, DTA analysis is still used to qualitatively evaluate the thermal stability of retained austenite in these heat treatable low alloy steels [35,36]. Therefore, the purpose of this study is to

investigate in detail the presence of finely distributed retained austenite microphase in a high-strength medium-silicon-bearing low-alloy micro-composite steel under various Q&B heat treatments. Various microstructural characterization methods including conventional light optical metallography, FE-SEM microscopy, XRD analysis, EBSD imaging, DTA and VSM analyses have been employed for both qualitative detection and quantitative estimation of retained austenite in contrast to other phases present in essentially bainite-martensite-retained austenite triple phase micro-composite microstructures.

## 2. Materials and Experimental Procedures

### 2.1. Primary Specimen Preparation

In this research work, a medium carbon commercial grade of DIN1.5025 steel sheet with 1 mm thickness and with chemical composition (in wt.%) of Fe-0.53C-1.67Si-0.72Mn-0.12Cr was used. The silicon and manganese contents of 1.67 and 0.72% were desirable in order to prevent carbide precipitation during isothermal transformation of austenite to bainite and hence to promote partitioning of carbon to the untransformed austenite, thus enabling stabilization of at least a fraction of carbon-enriched austenite down to room temperature. Besides, some untempered high-carbon martensite may also form in the micro-composite microstructures during final cooling depending on the stability of the austenite. To develop tough, strong steel with micro-composite microstructures containing bainite, martensite and retained austenite microconstituents, a suitable heat treatment schedule i.e., Q&B type process was conducted on the steel samples with different holding times. For this reason, the determination of critical temperatures of $Ac_1$, $Ac_3$, $M_s$ and $B_s$ was considered essential. Consequently, the critical temperatures of $Ac_1$, $Ac_3$ and $M_s$ for this steel sheet were determined by dilatometer measurements in a Gleeble 3800 thermomechanical simulator using heating and cooling rates of 0.2 °C/s and 150 °C/s, respectively. As suggested in reference [55], selection of very low heating rates is an important factor in determination of $Ac_1$ and $Ac_3$ near equilibrium conditions as well as applying high cooling rates is also required in determination of $M_s$ due to the prevention of austenite reversion from martensite. The corresponding temperatures were estimated as 765, 835 and 273 °C, respectively. In addition, the empirical equations, presented in Table 1, predicted the martensite ($M_s$) as well as bainite ($B_s$) start temperatures of 281 and 471 °C, respectively, showing also the accuracy of $M_s$ prediction. Accordingly, the Q&B heat treatment cycles were designed by combining quenching, isothermal nanobainitic ferrite formation and simultaneous carbon partitioning in order to achieve micro-composite microstructures containing various percentages of bainite, martensite and retained austenite microphases. Proposed heat-treated samples with dimensions of $30 \times 30 \times 1$ $mm^3$ were first normalized at 900 °C for 5 min in order to achieve a more homogenized starting microstructure in the samples. The subsequent Q&B type heat treatment cycles involved reaustenitizing at 900 °C for 5 min, quenching in a salt bath maintained at 350 °C (a temperature between $M_s$ and $B_s$) for times ranging from 5 s to 1 h, followed by water cooling to room temperature, as schematically depicted in Figure 1. In addition to these heat treatment cycles, some other samples in fully annealed conditions were also prepared in order to be able to compare the microstructural features with those of the micro-composite samples. The full annealing process included reaustenitizing the samples at 900 °C for 5 min followed by furnace cooling to room temperature.

**Table 1.** The empirical equations applied for calculating martensite ($M_s$) and bainite ($B_s$) start temperatures.

| Item | Empirical Equation | Calculated Values (°C) | Ref. |
|:---:|:---:|:---:|:---:|
| Martensite start temperature, $M_s$ | $M_s$ (°C) = 539 − 423C − 30.4Mn − 17.7Ni − 12.1Cr − 7.5Mo | 281 | [56] |
| Bainite start temperature, $B_s$ | $B_s$ (°C) = 656 − 58C − 35Mn − 75Si − 15Ni − 34Cr − 41Mo | 471 | [57] |

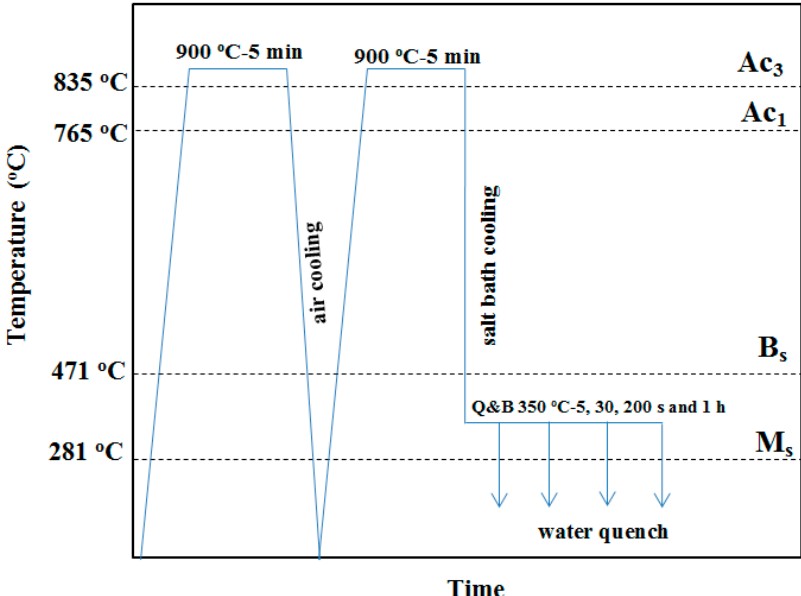

**Figure 1.** Schematic of Q&B heat treatment cycles carried out on sheet samples.

## 2.2. Microstructural Characterization and Analyzing

To reveal the microstructural features with sufficiently contrasting resolution, the heat-treated samples were cold mounted, ground and polished according to ASTM E3 standard and were etched using various chemical reagents as listed in Table 2. The LePera's etching reagent was successfully used for color metallographic observations in an OLYMPUS PMG3 light microscope (Olympus, Japan). For a better contrast and resolution, the samples were deeply etched in a 2% nital reagent and examined for microstructural details by scanning electron microscopy in an FE-SEM model TESCAN MIRA3 (Tescan, Brno, Czech Republic). Moreover, Clemex image analyzing software (Version 3.5.025) was used to analyze the optical images and measure the retained austenite volume fraction in at least five representative areas of $200 \times 200 \ \mu m^2$ at different locations of the samples and the average values were reported.

**Table 2.** Various chemical reagents used to reveal the microstructural features of Q&B heat-treated samples.

| Etching Reagent | Chemical Composition |
|---|---|
| 2% nital | 2 mL nitric acid, 98 mL ethanol |
| 5% picral | 5 g picric acid, 100 mL ethanol |
| Sodium metabisulfite | 7 g $Na_2S_2O_5$, 100 mL distilled water |
| Glyceregia | 9 mL glyceregia, 6 mL HCl, 3 mL nitric acid |
| LePera | 50 mL $Na_2S_2O_5$ 1% in aqueous solution, 50 mL picric acid 4% in ethanol |

The volume fraction and also the carbon content [58] of retained austenite were measured by XRD analysis (Rigaku SmartLab 9kW XRD) using Co-$K_\alpha$ radiation at 135 mA and 40 kV conditions. A step scan size within the 2θ interval angles of 45 to 130° at 7.2°/min rotating rate was performed. Both Xpert High Score (for phase identification) and Maud software programs were used for qualitative and quantitative investigations of retained austenite, respectively. The volume fraction of retained austenite was measured using a direct comparison approach, comparing the integrated intensities of (111), (200), (220), and (311) of FCC diffracted planes with (101), (002), (112) and (202) of BCC planes, respectively. Moreover, electron backscatter diffraction (EBSD) (Physical Electronics company, Chanhassen, MN, USA) technique was used with Scanning Auger Nanoprobe model Physical Electronics-PHI 710 EBSD system to detect different phases. The acceleration voltage was 15 kV with a beam current of 10 nA

and working distance of 8 mm. The angle between the surface normal of the samples with the horizon in front of the EBSD detector was 70° during the EBSD data acquisition from the scanned area of $50 \times 50 \ \mu m^2$ with the scanning step size of 150 nm. A TSL OIM7 software was subsequently used to analyze the EBSD acquisitioned data. For enhancing the map through data clean-up procedure, points with a Confidence Index (CI) of smaller than 0.2 were excluded.

The differential thermal analysis was another technique employed to investigate the thermal stability of retained austenite in various heat-treated samples. In this method, samples with a weight of 50 mg were thoroughly cleaned by an emery paper and then heated from ambient temperature to 700 °C in a Bahr dilatometer model DIL 801L at a heating rate of 10 °C/min under high purity argon gas atmosphere to prevent any oxidation. Furthermore, the samples with dimensions of $5 \times 5 \times 0.1 \ mm^3$ were prepared by using a wire-cut electric discharge machine (EDM) for room temperature magnetic measurements. These samples were characterized by the vibrating sample magnetometry (model MDKB of Danesh Pajouh Company, kashan, Iran) with a maximum field intensity of 1 Tesla.

## 3. Results and Discussion

### 3.1. Light and Electron Microscopic Observations

#### 3.1.1. Optical Micrographs

To reveal the microstructural features of the Q&B heat-treated micro-composite samples with sufficient contrast, various single and double light metallography techniques have been tried based on the chemical reagents involving nital, picral, sodium metabisulfite, glyceregia and LePera etching solutions, as listed in Table 2. To begin with, 2% nital and/or 5% picral etching reagents are widely used to reveal the microstructures of low alloy steels. Figure 2a,b show typical examples of optical micrographs recorded on samples etched with 2% nital and 5% picral solutions, respectively, showing the associated problems in revealing the microstructural constituents with reasonable contrast, commonly encountered with bainitic and martensitic samples. The etching reagents attack sharply the inclusions thus obscuring other features. Bainite plates appearing as slightly blackish or light brown features in the micrographs are still difficult to distinguish from martensite. The fine retained austenite islands, if any, cannot be revealed or distinguished from bainite and martensite packets in the micro-composite microstructures using this method.

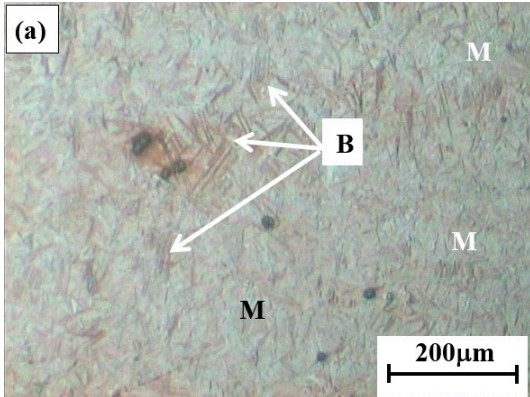 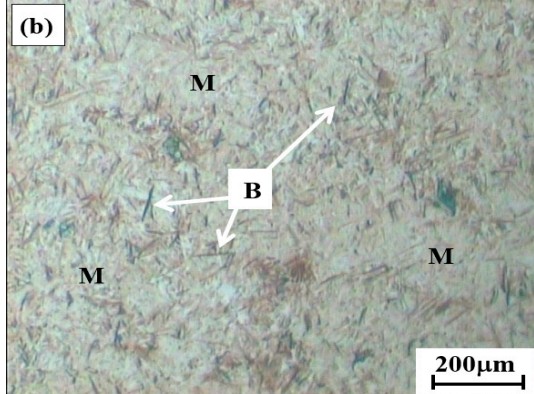

**Figure 2.** Optical micrographs of Q&B samples isothermally treated at 350 °C for 5 s after etching in: (**a**) 2% nital; and (**b**) 5% picral reagents. In these micrographs bainite plates (B) superimposed in the martensitic matrix (M) are observed by black or light black and light brown contrasting areas, respectively.

Color light metallography is normally considered to be a suitable method in order to detect fine grains of retained austenite amid other microconstituents in such microstructures. Typical light micrographs of the samples isothermally held at 350 °C in a salt bath for various holding times are

shown in Figure 3 following etching in LePera's reagent. Figure 3a presents the microstructure of samples partitioned for a short period of 5 s depicting clearly the shiny white islands of retained austenite with a good contrasting resolution from light brown martensitic matrix. The volume fraction of retained austenite islands was estimated as 3.6 vol.%. When the isothermal holding time was increased to 30 s, only a trace fraction of bainite formed, as is obvious from the dark colored plates forming at the prior austenite grain boundaries (Figure 3b), the rest being a mixture of martensite and some fine retained austenite grains. It is observed that the retained austenite is detectable as shiny white islands, in sharp contrast from light black plate-shaped bainite sheaves and brown colored martensitic matrix regions in this micrograph. In this heat-treated condition, the volume fraction of retained austenite is estimated as 3.4 vol.%, resulting in essentially a triple micro-composite microstructure consisting of martensite, a small fraction of bainitic ferrite and small islands of retained austenite. Further increase in isothermal holding time up to 200 s resulted in more bainite formation and as a consequence, extensive carbon partitioning during bainite formation into the adjoining prior austenite locations, as the presence of high silicon would prevent or at least delay formation of carbides despite the high carbon content in the steel (0.52%). As a result, the carbon-enriched untransformed austenite grains will have their local $M_s$ temperatures reduced significantly well below 350 °C on subsequent water-quenching, facilitating stabilization of at least a fraction of austenite down to room temperature, in addition to possible formation of some untempered, high carbon martensite. Figure 3c represents the microstructure of samples partitioned for 200 s at 350 °C in consistence with the above discussion showing the large brown colored regions of bainitic ferrite associated with superimposed discrete white islands of martensite/retained austenite (M/RA), thus developing a micro-composite microstructure. The overall phase mixture of martensite and retained austenite appeared as discrete islands of M/RA constituents with a combined fraction of about 43.2 vol.% estimated by point counting method of several micrographs, (Figure 3).

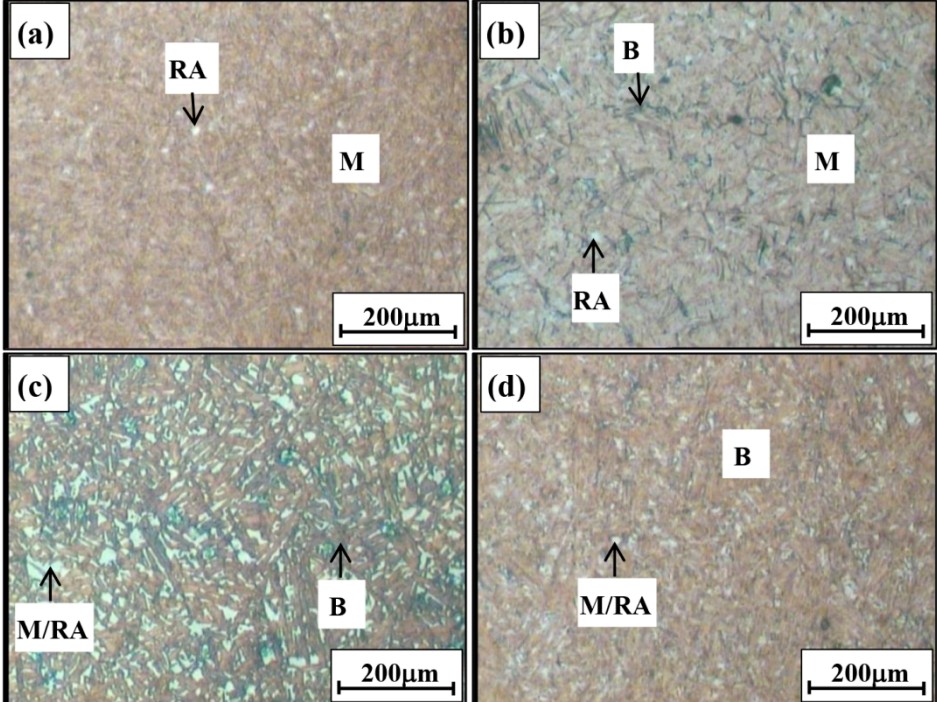

**Figure 3.** Light micrographs of Q&B samples isothermally treated at 350 °C molten salt bath for: (**a**) 5 s; (**b**) 30 s; (**c**) 200 s; and (**d**) 1 h after using LePera's etching reagent. The symbols of B, M, RA and M/RA stand for bainite, martensite, retained austenite and martensite/retained austenite microconstituents, respectively.

In addition, in the case of samples held for 1h at 350°C, it was not possible to distinguish retained austenite from martensite by optical microscopy even by using LePera's color etching. In effect, a large fraction of bainitic ferrite formed to the tune of 96.2 vol.% and only a small fraction of fine M/RA islands (about 3.8 vol.%,) could be seen (Figure 3d).

### 3.1.2. FE-SEM Micrographs

Figure 4 shows typical backscattered electron (BSE) micrographs of isothermally held samples at 350 °C for various durations in salt bath as stated above. Figure 4a reveals the microstructure of samples partitioned for short isothermal holding time of 5s, characterized with multivariant packets of martensitic crystals in a manner similar to that observed in light optical microscopy (Figure 3a), but contrary to the corresponding optical image, the retained austenite islands were not distinguishable by the scanning electron microscopy. By increasing partitioning time to 30 s (Figure 4b), a trace formation of slightly darker contrasting bainite plates can be observed in conjunction with the remaining multivariant packets of martensitic crystals, besides small islands of retained austenite. Under isothermal transformation conditions, bainite directly forms from austenite by nucleation and growth of bainitic ferrite. A further increase in isothermal holding time to 200 s resulted in marked bainite formation, appearing as grayish matrix phase area. The M/RA constituent islands contrasted in the same fashion and appeared dispersed together indistinguishably as blocky islands or thin films, emphasizing that neither was it possible to detect the retained austenite from martensite nor the volume fraction could be assessed from the viewpoint of electron microscopic observations. The reasonable explanation for this confusing result is that the martensite and retained austenite formed in the blocky islands or thin films had nearly similar chemical composition and so the BSE image contrast may not be able to distinguish between the two microphases. Granular bainite (GB), which consists of a highly dislocated irregular type of ferrite with second phases of a granular morphology, has also appeared in the microstructure, Figure 4c. The ferritic matrix comprises bunches of bainite, where thin retained austenite layers are present between the subgrains [59]. The structure of GB does not contain carbides. The mechanical properties of granular bainite are remarkably affected by the M/A islands. A higher bainite start temperature (Bs) promotes the formation of GB [60]. Figure 4d represents the electron micrograph for the samples isothermally partitioned for long holding time of 1h in the molten salt bath and it is comprised of somewhat light M/RA islands in a relatively featureless dark bainitic matrix. M/RA islands comprising essentially a mixture of martensite and retained austenite microconstituents appeared quite similar to those seen in the samples held for 200 s, emphasizing the inability of the BSE imaging technique to distinguish martensite and retained austenite decisively from each other. The presence of degenerated upper bainite that includes ferrite matrix with dispersed lath of retained austenite, has also been revealed in the microstructure. Moreover, in comparison with the associated color light micrograph shown in Figure 3a,b, it can be concluded that the retained austenite islands were observed more clearly by optical microscopy than by backscattered electron microscopy in an SEM. In contrast, the advantage of electron microscopic investigation is the possibility of achieving high magnification, which in turn enables sharply clear observation of bainite crystals appearing as dark contrasting sheaves and also multivariant martensitic packets in the same fashion as seen in the color light micrographs. These results indicate that the use of FE-SEM micrographs are not suitable for distinguishing retained austenite from martensite in M/RA microconstituents, even though a good contrasting distinction can be made between bainite and martensite packets. In contrast, the retained austenite islands have been clearly distinguished from martensite and bainite by using color metallography (Figure 3a,b).

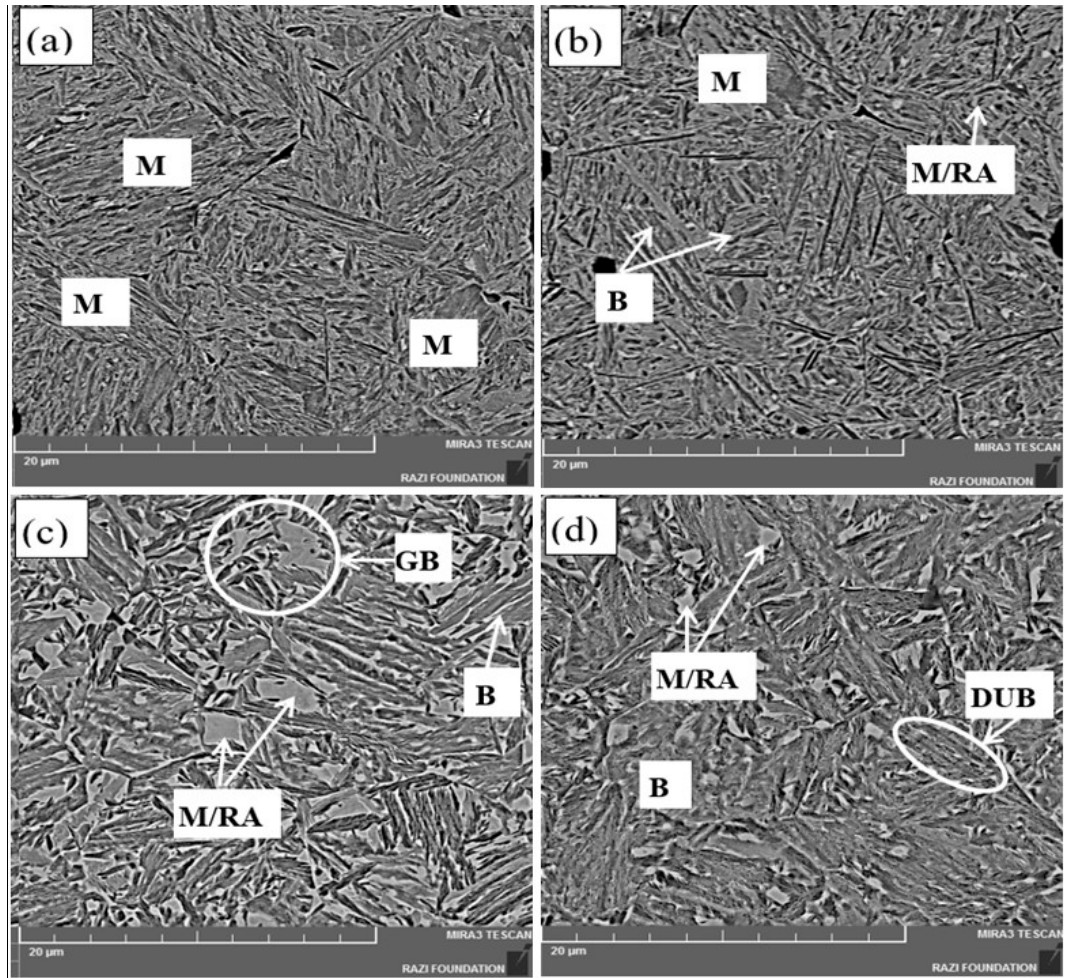

**Figure 4.** The backscattered electron micrographs of samples isothermally held at 350 °C for: (**a**) 5 s; (**b**) 30 s; (**c**) 200 s; (**d**) 1h, after following with room temperature water quenching.

### 3.1.3. EBSD Micrographs

Figure 5 represents typical EBSD phase maps (PMs) of samples partitioned at 350 °C for various isothermal holding times. In these maps, the FCC retained austenite phase (green colored areas) is clearly distinguishable from all of the remaining red contrasted BCC bainite and/or martensite phases. From this viewpoint, the micro-composite microstructures of the Q&B samples consist essentially of a mixture of just FCC and BCC microconstituents. Carbide precipitation, if any, cannot be revealed because of the limited resolution of EBSD (about 0.1 µm). Moreover, both the bainite and martensite microphases have similarities in respect of phase transformation mechanisms as well as crystallographic orientation in relation to the parent austenite phase.

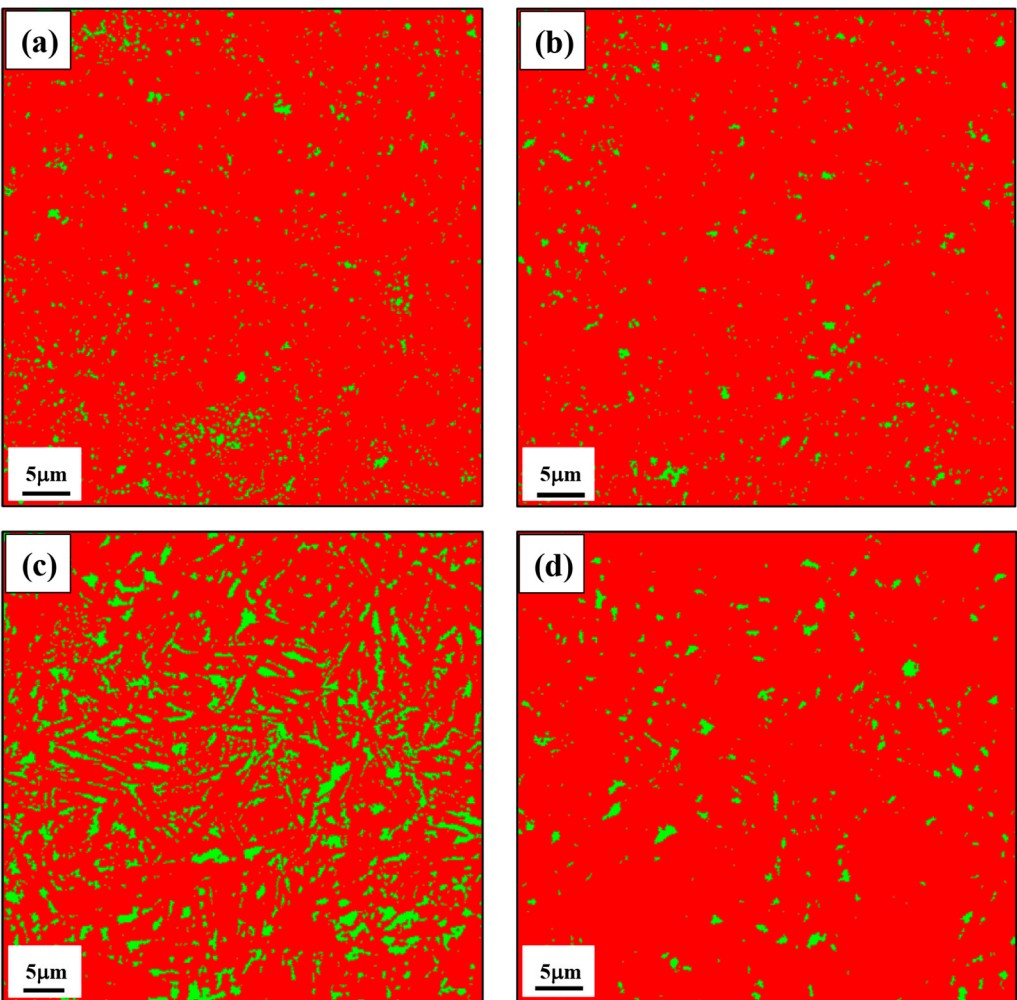

**Figure 5.** EBSD images of PMs for the samples held isothermally at 350 °C salt bath for: (**a**) 5 s; (**b**) 30 s; (**c**) 200 s; (**d**) 1 h; and then followed by subsequent quenching in water. The green regions are FCC retained austenite and the red ones are BCC bainite and/or martensite. The bainite and martensite regions cannot be detected from each other in these electron micrographs.

For more information, the PMs have been also used to measure the volume fractions of carbon-enriched austenite retained in various Q&B heat-treated samples and the concerned data is reproduced in Table 3. The maximum volume fraction of retained austenite was estimated to be about 15.3 (±1) vol.% in the samples corresponding to moderate isothermal holding time of 200 s. A longer holding of 1h at 350 °C caused further decomposition of austenite to bainite and subsequently very little untransformed austenite existed that appeared as M/RA islands during final quenching to room temperature. For this reason, the amount of retained austenite has largely decreased from 15.3 to 3.1 (±1) vol.% with the increase in isothermal holding time from 200 s to 1 h. In summary, a good conclusion can be drawn by comparing the retained austenite data collection by both the EBSD as well as light color metallography methods, as the volume fractions of retained austenite measured by the two techniques are nearly the same. For instance, the low level of 3.1 (±1) vol.% retained austenite in the Q&B sample held for 1 h, estimated by the EBSD technique, could also be measured with fairly good accuracy by light color metallography, even though both the techniques cannot detect very fine austenite grains due to limitation of resolution.

**Table 3.** The variation in volume fractions of retained austenite taken from Q&B heat-treated samples determined by different analyzing methods.

| Q&B Samples | Volume Fractions of Retained Austenite (vol.%) | | |
|:---:|:---:|:---:|:---:|
| | Optical Microscopy (±4) | XRD (±1) | BSD (±2) |
| 350 °C-5 s | 3.6 | 3.1 | 3.1 |
| 350 °C-30 s | 3.4 | 4.5 | 4.4 |
| 350 °C-200 s | - | 18.0 | 15.3 |
| 350 °C-1 h | - | 6.9 | 3.1 |

It is noteworthy that though the EBSD PMs shown in Figure 5 can be used to determine the distribution and morphological features of retained austenite in the micro-composite microstructures, the application of a combination of image quality (IQ) maps with PMs shown in Figure 6 can provide a detailed information about the location, distribution and morphological features of various microconstituents in the microstructures. The only difference between PM and IQ + PM mapping lies in the fact that it is not possible to distinguish bainite from martensite in the PM micrographs, while it can be observed with nice contrasting resolution from IQ + PM images (Figure 6). In other words, the PMs presented in Figure 5 show that the entire BCC matrix phase microstructure corresponding to all of the micro-composite phases is just red colored, and it is not possible to distinguish bainite from martensite in the microstructures. However, the IQ + PM electron micrographs shown in Figure 6 indicate that the bainitic areas appear as bright red regions in the IQ + PM images due to the higher confidence indexing of bainitic crystals. In turn, the fresh martensitic areas have poor indexing of crystallographic intensity than that of bainitic regions due to the higher intensity noise of internal strains and dislocations. As a consequence, the bainitic crystals are generally much better indexed in EBSD images than that of martensite ones and hence the martensitic areas appear as darker contrasting (somewhat poorly indexed) regions in the IQ + PM maps. Moreover, considering these maps indicate that by increasing holding time from 5 s to 1 h, the amount of bainite (bright red regions) increased significantly and as a consequence, the fraction of martensite formed during final cooling decreased appreciably (Figures 5 and 6). Secondly, the morphology of retained austenite regions are mainly thin films or discrete blocky-shaped islands which are located and distributed mostly between the packets of martensite and bainite crystals. This morphological feature of retained austenite can be associated with the partitioning of carbon atoms from bainite to the adjacent prior austenite areas. When the amount of retained austenite stabilized at room temperature was recorded maximum at about 15.3 (±1) vol.% in the Q&B samples held for a moderate time of 200 s at 350 °C salt bath (Figures 5 and 6, Table 3), more carbon atoms have been rejected from bainite into the adjacent prior austenite areas. As a result, a fraction of films and pools of retained austenite which were located between bainitic crystals were stabilized down to room temperature on the subsequent water quenching, resulting in the retention of a high fraction of carbon-enriched stable austenite in the micro-composite microstructures with mostly thin film-like or blocky morphologies [61].

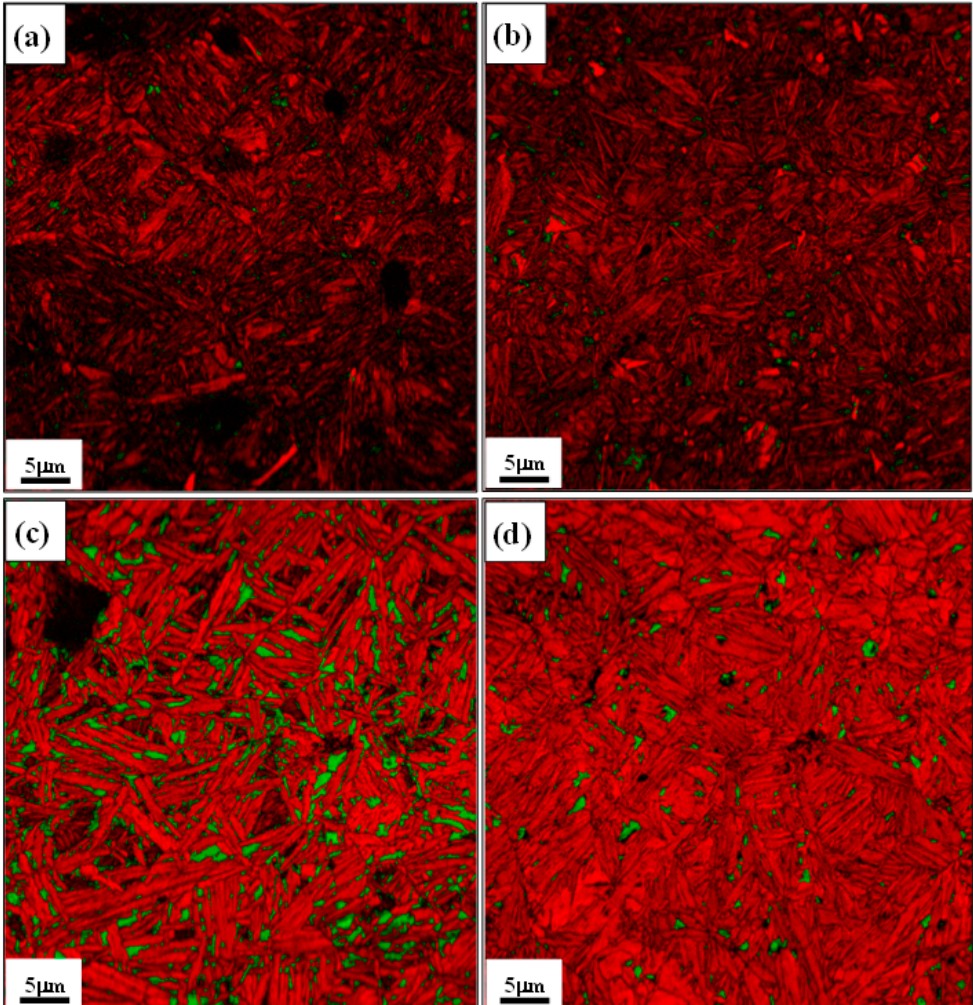

**Figure 6.** The combination of IQ and PMs electron images obtained by EBSD for the samples isothermally treated at 350 °C for: (**a**) 5 s; (**b**) 30 s; (**c**) 200 s; (**d**) 1 h; and then followed by subsequent water quenching. The green, black and bright red colored regions are retained austenite, martensite and bainitic ferrite microphases, respectively.

### 3.2. Thermal Stability of Retained Austenite

The thermal stability of retained austenite is an important factor for post-Q&B heat treatment processes such as colorizing and galvanizing. This is due to the fact that the retained austenite can decompose and consequently local expansion can lead to variation of size and shape in actual industrial applications, if they are heated above the temperatures of stability during component fabrication. In this work, the DTA thermal technique was used in order to determine the presence as well as thermal stability of retained austenite in various Q&B heat-treated samples. The fully annealed and Q&B heat-treated specimens with weights of nearly 50 ± 2 mg were heated from room temperature to 700 °C at a linear heating rate of 10 °C/min, and the typical thermal curves obtained in these experiments are reproduced in Figure 7. It can be observed that there is an exothermic peak in each of the DTA curves of the Q&B heat-treated samples, whereas no peaks have been detected in the DTA curves obtained from fully annealed ones. Moreover, the DTA curves showed that these exothermic peaks appeared in the temperature range 139 to 285 °C, which must be related to the decomposition of retained austenite. The variation of retained austenite decomposition temperature from 139 to 285 °C can be associated with the variation of retained austenite volume fraction as well as retained austenite carbon concentration enriched during isothermal bainitic holding at 350 °C. This is why no peaks were found in the DTA curves of fully annealed samples because the corresponding microstructural

constituents comprised simply a mixture of ferrite and cementite microconstituents. From this point of view, it is obvious that the DTA thermal analysis can be successfully used to qualitatively determine the presence of retained austenite in the Q&B samples.

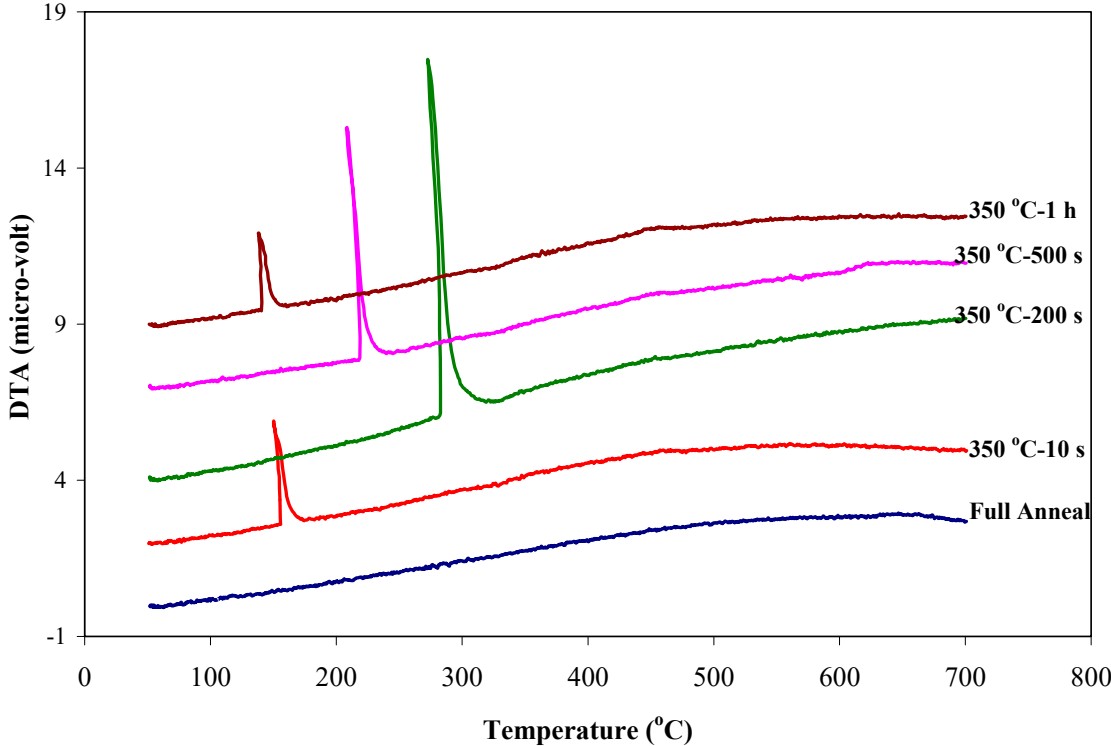

**Figure 7.** The DTA thermal curves taken from various Q&B heat-treated samples at 350 °C salt bath for different holding times in comparison to that of full annealed ones.

For more information, Table 4 summarizes the DTA thermal stability data of retained austenite in conjunction with the associated XRD measurements for different Q&B heat-treated samples. Firstly, it is evident that as the isothermal bainitic holding time increased from 10 to 200 s, the DTA decomposition temperature of retained austenite increased from 151 to 285 °C and then surprisingly decreased to a lower value of 139 °C with longer holding time of 1 h. Secondly the DTA decomposition peak intensities showed a similar trend, i.e., with an increase in holding time from 10 to 200 s, the peak intensity increased from 3.2 to 11.2 μV and then decreased to 2.5 μV for the longest duration time of 1 h. This abnormal trending of thermal stability for retained austenite can be explained by the variation of its volume fraction and carbon concentration measured by the XRD analysis, as given below. On one hand, as the amount of retained austenite increased from 4.8 to 18 vol.% and then decreased to 6.9 vol.% for bainitic holding times of 10, 200 s and 1 h, respectively, it is expected that the intensity of retained austenite decomposition peak obeys the same trending variation. On the other hand, as the bainite transformation proceeds at 350 °C for holding times of 10 to 200 s, carbon content of retained austenite increases from 0.57 to 1.12 wt% resulting in higher thermal stability of retained austenite up to 285 °C and then surprisingly it decreases to 139 °C corresponding to the higher carbon content of 1.34 wt% for Q&B sample heat-treated for 1 h at 350 °C. A possible explanation for this lowering of retained austenite decomposition temperature with 1.34 wt% carbon concentration can be made according to the solubility limit of carbon in retained austenite. It is clear that the potential level of soluble carbon content of retained austenite is variable in this low alloy steel as can be developed during isothermal holding at austenitizing temperature of 900 °C. For this reason, from a portion of Fe-C equilibrium phase diagram shown in Figure 8, it can be claimed that the solubility limit of carbon in prior austenite can be varied from 0.53 to 1.26 wt% emphasizing that the maximum amount of solute carbon in austenite at 900 °C is nearly 1.26 wt% in a binary Fe-C system, and this amount can

even decrease to somewhat lower than 1.26 wt% in the investigated low alloy steel. The presence of 0.72 wt% Mn and 1.67 wt% Si in this low alloy steel acts as a source of lattice distortion for retained austenite unit cells, suggesting that the maximum solubility limit of carbon in retained austenite is lower than that of 1.26 wt% and some of related austenite areas are supersaturated in relation to solute carbon atoms and hence, become more sensitive to decomposition on the subsequent DTA heating cycles. This result is consistent with our experimental work indicating that the maximum DTA peak temperature of 285 °C for thermal stability of retained austenite occurred in the moderate time of 200 s treated Q&B samples corresponding to the moderate XRD carbon content of 1.12 wt%, respectively.

**Table 4.** The summarized data of DTA for retained austenite decomposition temperature and its peak intensity in conjunction with the associated volume fraction and carbon concentration of various heat-treated samples.

| Q&B Samples | DTA Data | | XRD Measurements | |
|---|---|---|---|---|
| | RA Decomposition Temperature (°C) | RA Peak Intensity (Micro-volts) | RA Volume Fraction (vol.%) | RA Carbon Content (wt%) |
| Q&B 350 °C-10 s | 151 | 3.2 | 4.8 | 0.57 |
| Q&B 350 °C-200 s | 285 | 11.2 | 18 | 1.12 |
| Q&B 350 °C-500 s | 222 | 7.4 | 13.4 | 1.29 |
| Q&B 350 °C-1 h | 139 | 2.5 | 6.9 | 1.34 |

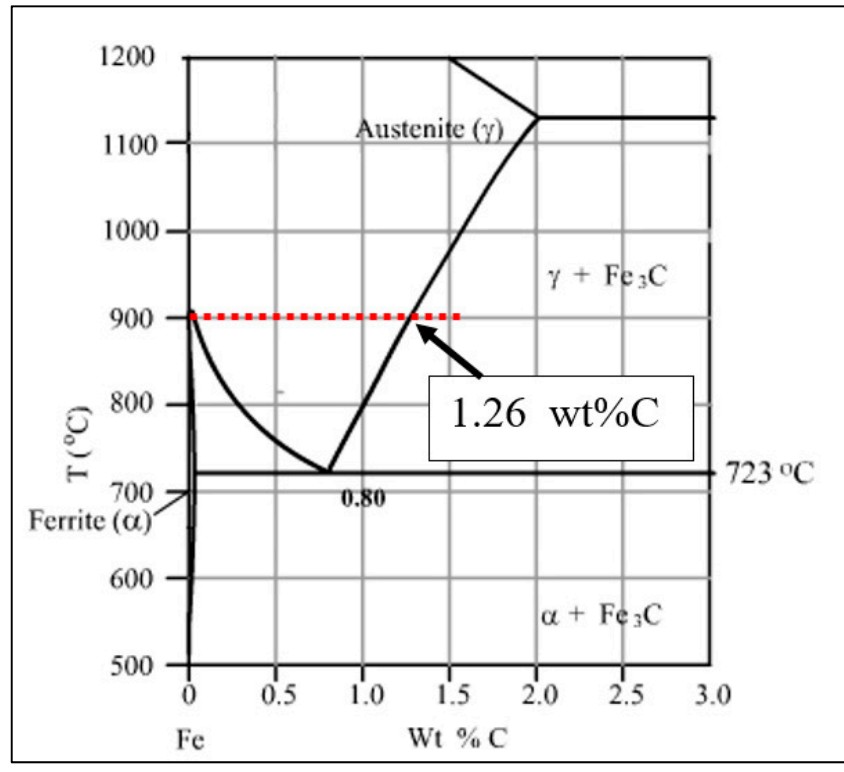

**Figure 8.** A portion of binary Fe-C equilibrium phase diagram indicating the maximum solubility limit of carbon in austenite is 1.26 wt% at 900 °C.

Another possible explanation for further supporting of these arguments can be made according to the fact that a higher density of vacancy can be developed during rapid salt bath cooling from 900 °C austenitizing temperature to 350 °C bainitic heat treatment condition. It is obvious that the density of vacancies in a crystalline metallic material can be exponentially increased with increasing temperature up to the level of about 1 over 10,000 atomic position at temperatures close to that of melting point. For this reason, as the Q&B samples are austenitized at 900 °C, the level of vacancy

concentration increases in the austenite and so the potential for solubility of interstitial carbon atoms can increase up to the value lower than that of 1.26 wt% according to the binary equilibrium Fe-C phase diagram shown in Figure 8. On the subsequent rapid 350 °C salt bath cooling, the high level of vacancy concentration can be developed in the austenite and so this austenite has the potential to accommodate the same level of carbon concentration as the 1.26 wt% during 350 °C isothermal bainitic holding time.

In summary, it can be concluded that the variation of decomposition temperature for retained austenite in the Q&B heat-treated samples can be related to the variation of solute carbon concentration in retained austenite. Initially, with increasing isothermal holding time in the molten salt bath up to 200 s, the decomposition temperature increases from 151 to 285 °C due to the higher thermal stability of retained austenite formed until this time. The reason for this stability is that with increasing isothermal holding time, more carbon atoms diffuse from the primary supersaturated bainitic crystals into the adjacent prior austenite areas, lowering diffusion of carbon and iron atoms and so causing higher thermal stability of retained austenite on heating. In contrast, on increasing isothermal bainitic holding time from 200 s to 1 h, the decomposition temperature of retained austenite decreases from 285 to 139 °C, respectively. This abnormal phenomenon can be supported by solubility limit of carbon atoms in the retained austenite. It is clear that as a consequence of more bainite formation during prolonged 350 °C bainitic holding time, more carbon atoms diffuse into the adjacent prior austenite areas, and so more carbon concentration can be developed in the retained austenite areas. Therefore, the bainite formation acts as a source for more partitioning of carbon atoms to the prior austenite which results in the associated higher supersaturated retained austenite areas in relation to carbon concentration. Hence, the tendency to form carbide and ferrite during reheating of the higher supersaturated retained austenite phase gets enhanced and the retained austenite tends to decompose easily and faster at lower temperatures, resulting in a decreasing trend in the decomposition temperature from 285 to 139 °C with higher isothermal holding time from 200 s to 1 h in the Q&B heat-treated samples, respectively. It is worth mentioning that these thermal stability results of retained austenite are consistent with the DSC thermal data, which have been previously reported by Moor et al. [29] and Shi et al. [35], who investigated the thermal stability of retained austenite in TRIP and Q&P heat-treated steels, respectively. Their studies proved that the presence of more than 1.5% Si with a sufficient amount of carbon concentration prevented retained austenite decomposition even at a relatively high temperature range of 200 to 500 °C.

### 3.3. Magnetic Analysis

A vibrating sample magnetometer (VSM) is another device used in the present study to investigate the presence of retained austenite in the isothermally partitioned micro-composite samples. Figure 9 shows the curves of variation of magnetization versus applied magnetic fields for fully annealed samples in comparison with that of Q&B heat-treated ones at 350 °C for 200 s. For a better comparison, the extracted saturation magnetization ($M_s$) and coercivity ($H_c$) data recorded from these magnetic curves are reproduced in Table 5, indicating that the saturation magnetization of fully annealed samples is much higher than that of Q&B ones. In turn, the coercivity of Q&B heat-treated samples is much higher than that of fully annealed ones. These results can be rationalized to the presence of non-magnetic FCC retained austenite microphase developed in the Q&B micro-composite microstructures in comparison to that of full annealed samples containing just carbide and ferrite microphases.

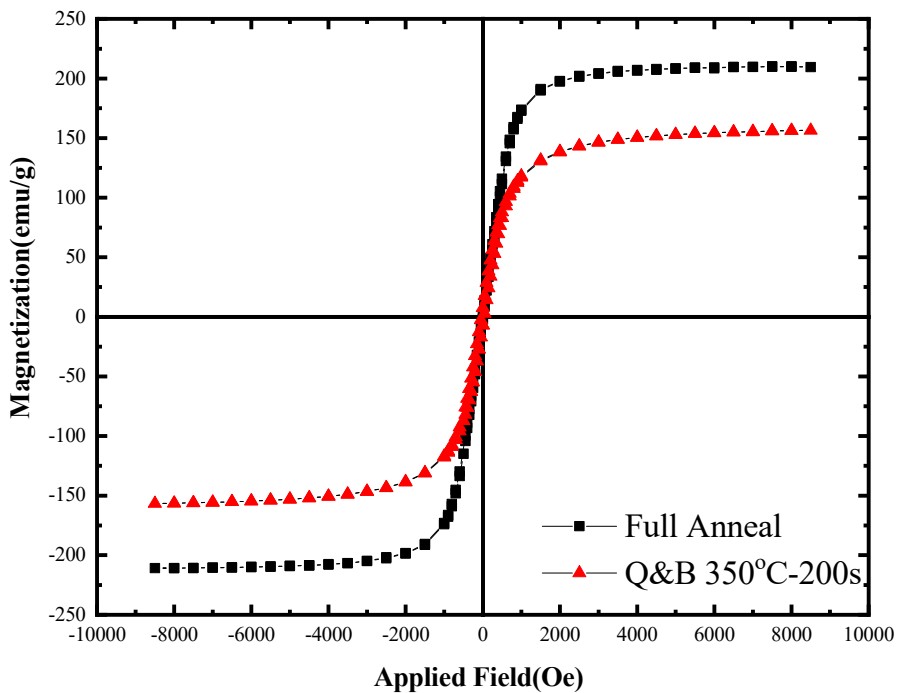

**Figure 9.** The comparison of VSM curves showing the variation of magnetization versus applied field in the full annealed and Q&B isothermally heat-treated samples at 350 °C for 200 s.

**Table 5.** The variation of magnetic parameters extracted from the VSM curves shown in Figure 8.

| Heat-Treated Sample | Saturation Magnetization (Ms) (emu/g) | Coercivity (Hc) |
|---|---|---|
| Full annealed | 210.18 | 9.27 |
| Q&B at 350 °C-200 s | 156.59 | 35.68 |

### 3.4. XRD Confirmations

Typical XRD patterns of the samples partitioned at 350 °C salt bath for isothermal holding times of 200 s and 1 h are shown in Figure 10. As can be seen, the patterns involve only two sets of FCC and BCC diffracted planes. The peaks of (111), (200), (220) and (311) planes which appeared with lower intensity in 2θ angles of 51°, 60°, 90° and 112° correspond to FCC retained austenite, while all the other peaks correspond to either bainite or martensite. The multiphase microcomponents of Q&B heat-treated samples associated with the XRD patterns are classified into the BCC bainitic ferrite and/or martensite with FCC retained austenite microphases. No carbide precipitation has occurred in the either bainite or martensite microphases confirming development of a triple micro-composite microstructural formation involving bainite-martensite-retained austenite microphases. The XRD analysis is consistent to the micro-composite microstructures colored with both of the EBSD and light metallography methods, confirming the results of DTA thermal stability and magnetic measurements of retained austenite as well.

For more information, the volume fraction of retained austenite has been calculated by applying the Maud software using the XRD patterns and the data are included in Table 3 in comparison with the associated light color metallography and EBSD data. As can be seen, firstly the volume fractions of retained austenite measured with XRD analysis have increased from 3.1 to 18.0 (±2) vol.% with increasing partitioning from 5 to 200 s and then decreased to 6.9 (±2) vol.% for 1h isothermal holding time, respectively. Secondly, although the trend of the variation of retained austenite volume fraction measured by XRD is consistent with the trends observed with both the light metallography and EBSD data, the related XRD measurements show somewhat higher retained austenite fractions than those made by optical and EBSD techniques. The reason for these slightly higher XRD estimates of retained

austenite is because of the advantage in respect of the detection and data acquisition of diffracted X-rays even from very fine grains of retained austenite at nanometer level in comparison to relatively large value of investigated 150 nm step size for EBSD analysis and long light wavelength for optical metallography examination. These XRD results are consistent with the literature reported accuracy of XRD data to about 0.5–1 vol.% of retained austenite detection in the samples with near random crystallographic orientation [48,53].

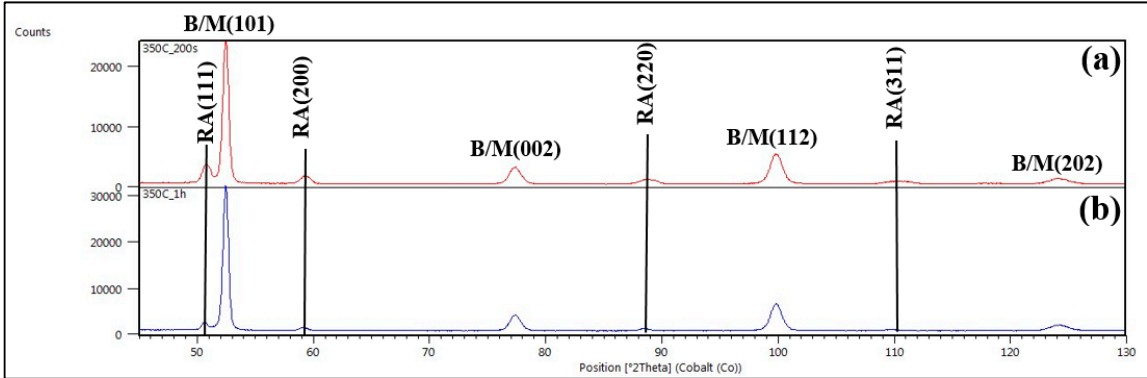

**Figure 10.** The XRD patterns of Q&B samples isothermally heat-treated at 350 °C for: (**a**) 200 s; and (**b**) 1 h. The locations of FCC retained austenite (RA) peaks are highlighted with solid lines, while the BCC bainite/martensite indices are shown with B/M symbols, respectively.

## 4. Conclusions

In this study, the sheet samples of a medium carbon Si-bearing low alloy grade of DIN1.5025 steel with 1 mm thickness were subjected to Q&B type heat treatment comprising austenitization at 900 °C for 5 min, quenching to a temperature 350 °C in the bainitic range in a molten salt bath slightly above the $M_s$ temperature (281 °C), holding for durations ranging between 5 s to 1 h, followed by water cooling. The aim of the study was to understand the phase transformation behavior including carbon partitioning and austenite stabilization during Q&B treatment in order to be able to design suitable heat treatment for developing micro-composite microstructures consisting of bainite, martensite and retained austenite microconstituents. This in turn would impart high strength with adequate ductility and strain hardening capacity in the steel. The following conclusions can be drawn:

The conventional light metallography technique using different etching reagents such as nital, picral, glyceregia and sodium metabisulfite solutions were not successful to detect the retained austenite from bainite and martensite, while a good contrasting resolution was made between these microphases by using LePera's etchant.

1. The LePera's etching reagent was used to reveal bainite, martensite and retained austenite by colorizing these microphases in the microstructures of Q&B samples isothermally heat-treated at 350 °C for shorter than 30 s. In holding times longer than 30 s, bainite was still distinguished from other microphases, but it was not possible to make a good contrasting resolution between martensite and retained austenite formed in the M/RA as they appeared as white blocky discrete islands or thin films.

2. In comparison to the possibility of identifying different phases by color light metallography, the FE-SEM microscopy was unable to clearly detect different microphases from each other. However, it could reveal the bainite morphologies including granular and/or degenerate upper bainite in the microstructures.

3. The EBSD analysis was able to reveal the FCC retained austenite from rest of the BCC bainite/martensite microconstituents. In addition, the volume fraction, morphology and distribution of retained austenite were clearly determined by this method. The volume fraction of retained austenite increased from 3.1 to 15.3 vol.% with increase in isothermal holding time

from 5 to 200 s. However, it decreased to 3.1 vol.% following isothermal holding for 1h due to extensive austenite decomposition and carbon redistribution from bainite into the adjacent prior austenite locations. The retained austenite appeared as blocky grains and thin film morphologies in the micro-composite microstructures.

4. The volume fractions of retained austenite measured by EBSD technique were largely in accordance with those obtained by color light metallography and XRD. The quantitative measurements of retained austenite performed by Rietveld process using Maud software were slightly higher than the values measured by both EBSD and color light metallography methods apparently due to the ability of the technique for getting diffraction even from finely divided retained austenite.

5. The DTA technique was used to determine the thermal stability of retained austenite in the micro-composite samples. The exothermic peaks on the DTA curves occurred in the temperature range of 139 to 285 °C and this range of decomposition temperature was attributed to the retained austenite decomposition and carbide formation.

6. The carbon concentration of retained austenite was continuously increased from 0.57 to 1.34 wt% with increasing isothermal bainitic holding time from 1s to 1h, respectively. The higher carbon concentration of retained austenite with longer isothermal bainitic holding time was related to more redistribution of carbon from bainite to the adjacent prior austenite areas developed with the progress of bainite formation on longer isothermal holding.

7. The thermal stability of retained austenite was justified with an abnormal trending by increasing isothermal bainitic holding times from 10s to 1h. The decomposition temperature of austenite was continuously increased from 151 to 285 °C and then surprisingly decreased to 139 °C with increasing isothermal bainitic holding times from 10 to 200 s and then to 1h, respectively. This abnormal trending of thermal stability for retained austenite was related to its level of solute carbon concentration.

8. The magnetic measurement was successfully carried out to qualitatively detect retained austenite in the Q&B heat-treated micro-composite samples by measuring the saturation magnetization and coercivity in comparison to the associated values taken from full annealed samples.

**Author Contributions:** The contribution of authors can be explained as below according to CRediT Taxonomyof contributor roles. Conceptualization: H.R.K.Z., S.S.G.B. and M.C.S.; Methodology: H.R.K.Z., S.S.G.B. and M.C.S.; Software: S.P.; Validation: S.P.; Formal analysis: S.P. and H.R.K.Z.; Investigation: S.P.; Resources: S.P., H.R.K.Z., S.S.G.B. and M.C.S.; Data curation: S.P. and H.R.K.Z.; Writing—original draft preparation: S.P. and H.R.K.Z.; Writing—review and editing: S.S.G.B. and M.C.S.; Visualization: S.P. and H.R.K.Z.; Supervision: S.S.G.B.; Project administration: S.S.G.B.; Funding acquisition: H.R.K.Z. and S.S.G.B. and M.C.S.

**Funding:** This research received no external funding.

**Conflicts of Interest:** The authors declare no conflict of interest.

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
