# Peer review of "Detection and Estimation of Retained Austenite in a High Strength Si-Bearing Bainite-Martensite-Retained Austenite Micro-Composite Steel after Quenching and Bainitic Holding (Q&B)"

_metals, doi:10.3390/met9050492_

Round 1

Reviewer 1 Report

The authors have applied a TRIP type annealing cycle to a medium C-Mn-Si strip steel and studied the evolution of the resulting microstructure as a function of bainitic holding time using multiple techniques (OM, SEM, EBSD, XRD, DTA, VSM). While there are some deficiencies in the experimental method - the optical microscopy images provided are of low resolution and appear to be out of focus and the XRD analysis does not include of the retained austenite carbon content, the work seems to be competently done and the results are believable. The main problem is that there is absolutely nothing new, original or unexpected in the manuscript. It would make a very useful teaching aid but this reviewer finds it difficult to see how it advances the field. If the authors could expand the DTA approach (which is not commonly applied to this type of study) by correlating the austenite decomposition peak temperature and intensity with the RA fraction and carbon content from XRD and the RA island size/morphology from EBSD  then this could provide a original contribution.

Author Response

Answer to comments of reviewer 1

Comment 1: The authors have applied a TRIP type annealing cycle to a medium C-Mn-Si strip steel and studied the evolution of the resulting microstructure as a function of bainitic holding time using multiple techniques (OM, SEM, EBSD, XRD, DTA, VSM). While there are some deficiencies in the experimental method - the optical microscopy images provided are of low resolution and appear to be out of focus and the XRD analysis does not include of the retained austenite carbon content, the work seems to be competently done and the results are believable. The main problem is that there is absolutely nothing new, original or unexpected in the manuscript. It would make a very useful teaching aid but this reviewer finds it difficult to see how it advances the field. If the authors could expand the DTA approach (which is not commonly applied to this type of study) by correlating the austenite decomposition peak temperature and intensity with the RA fraction and carbon content from XRD and the RA island size/morphology from EBSD then this could provide a original contribution.

Among the recorded optical microscopy images available, only the relatively good micrographs have been selected for presentation in figures 2 and 3 of the manuscript.

The carbon contents of the retained austenite measured through the analysis of the XRD spectra have been included in Table 4 and later discussed in section 3.2.

The DTA approach has been expanded by correlating the austenite decomposition peak temperature and intensity with the RA fraction and carbon content measured through the XRD analysis. Please read the section 3.2. thermal stabilities of retained austenite and the associated conclusions numbered 7 and 8 as well.

Reviewer 2 Report

1.     Clearly state the novelty aspects of the paper;

2.     More details referring EBSD are necessary. What about post-processing of EBSD data? What kind of clean-up procedures were applied? What size of area for each sample was investigated? Which values of grain tolerance angle and minimum grain size were applied?

3.     The quality of Fig. 2 and 3 is poor. It has to be improved. Moreover please, mark the identified constituets in Fig 2.

4.     What kind of method for retained austenite estimation from light microscopy images was employed? What about area of measurements? These informations have to added to the section 2.2.

5.     BSE detector is not the best one to observe the multi-phase structures. For this purpose better is SE detector. Despite this, the structures in Fig. 4 allow for identification the different types of bainite. One can see at least granular bainite and  degenerated upper bainite. Please mark them in Fig 4c and 4d. The description of these morphologies should be added to the text. It can be combined with EBSD results. In reference to this please modify the point 3 in Conclusions.

6.     Please correct term „BSD” on  EBSD in Table 3.

7.     In Conclusions please underline the new findings of the paper compared to the known knowledge.

Author Response

Answer to comments of reviewer 2

Comment 1:Clearly state the novelty aspects of the paper;

In this study, different qualitative and quantitative characterization techniques were used for detecting and estimating retained austenite in the micro-composite bainite-martensite-retained austenite microstructures of Q&B heat treated high-silicon low-alloy medium-carbon DIN1.5025 steel sheets, besides the comparison of different techniques in the identification and comparison of different phases. In addition, we also used the differential thermal analysis (DTA) in order to innovatively check the stability of retained austenite in the microcomposite microstructures by considering the retained austenite decomposition peaks. Furthermore, we have tried to understand and discuss the relationship between the thermal stability of retained austenite determined by DTA with the corresponding XRD measurements for volume fraction and the carbon concentration (section 3.2. Thermal stabilities of retained austenite). The conclusions # 7 and 8 summarize the salient results.

Comment 2:More details referring EBSD are necessary. What about post-processing of EBSD data? What kind of clean-up procedures were applied? What size of area for each sample was investigated? Which values of grain tolerance angle and minimum grain size were applied?

The following additional details for the EBSD technique have been added in  section 2.2.

Data clean-up procedure: Points with a Confidence Index (CI) of smaller than 0.2 were  excluded

The angle between the samples’ surface normal and the horizon in front of the EBSD detector: 70°

Size of the analyzed area: 50 × 50 μm2

Step size: 150 nm

Data collection and analyzing software: TSL OIM 7

Comment 3: The quality of Fig. 2 and 3 is poor. It has to be improved. Moreover please, mark the identified constituets in Fig 2.

As stated above, among the optical microscopy images recorded on the samples, only the relatively good micrographs have been selected for presentation in figures 2 and 3 of the manuscript. In addition, the microconstituents in images of Fig. 2 have been labeled.

Comment 4:What kind of method for retained austenite estimation from light microscopy images was employed? What about area of measurements? These informations have to added to the section 2.2.

The Clemex image analyzing software was used to analyze the light microsopy images and estimate the retained austenite volume fraction in at least five representative areas of 200×200μm2 taken at different locations of the specimens and the average values were reported.

The above-mentioned information has been added to section 2.2. of the manuscript.

Comment 5: BSE detector is not the best one to observe the multi-phase structures. For this purpose better is SE detector. Despite this, the structures in Fig. 4 allow for identification the different types of bainite. One can see at least granular bainite and  degenerated upper bainite. Please mark them in Fig 4c and 4d. The description of these morphologies should be added to the text. It can be combined with EBSD results. In reference to this please modify the point 3 in Conclusions.

Although almost all researches use SE detector to take SEM images for multi-phase microstructures but, in some cases  BSE detector sometimes seems to provide better identification of microconstituents present in the microstructures with the possibility of atomic number contrast (if phase compositions are different) and improved depth of field. For example, please consider the electron images given below. The images on the left and the right sides are taken from the same position by SE and BSE detectors, respectively, showing better contrast with the BSE detector.

In addition, the latter comment about marking and/or identification of different bainite morphologies has been addressed and necessary modifications made in the text.

Comment 6: Please correct term „BSD” on  EBSD in Table 3.

The term has been corrected in Table 3.

Comment 7: In Conclusions please underline the new findings of the paper compared to the known knowledge.

As stated above, different microstructural characterization techniques have been employed in this study for both qualitative as well as quantitative metallography and phase estimation, in particular to detect and estimate the retained austenite in the microcomposite bainite-martensite-retained austenite microstructures of Q&B treated DIN1.5025 steel sheets. A comprehensive comparison of different techniques revealed various possibilities and limitations. Moreover, we used the differential thermal analysis (DTA) for the first time to check the presence of retained austenite in the microcomposite microstructures by considering the retained austenite decomposition peaks. Furthermore, we have discussed the relationship between the thermal stability of retained austenite determined by DTA with the corresponding XRD measurements for volume fraction and the carbon concentration. The same has been illustrated in section of 3.2 and later summarized in conclusions 

Reviewer 3 Report

The article in question titled " Detection and Estimation of Retained Austenite in a 3 High Strength Si-Bearing Bainite-Martensite4 Retained Austenite Micro-composite Steel after 5 Quenching and Bainitic Holding (Q&B)" adreses all the research questions defined in the abstract. The manuscript is written with a clear language. 

Before the acceptance of the current  manuscript the following question needs to be clarified. 

1- Using the methods described in  the manuscript is it possible to have a conclusion about the effect of amount of initial retained austenite level to the strenght and ductility of the steel ?

Author Response

Answer to comments of reviewer 3

Comment 1:The article in question titled " Detection and Estimation of Retained Austenite in a 3 High Strength Si-Bearing Bainite-Martensite4 Retained Austenite Micro-composite Steel after 5 Quenching and Bainitic Holding (Q&B)" adreses all the research questions defined in the abstract. The manuscript is written with a clear language. Before the acceptance of the current  manuscript the following question needs to be clarified. 

1- Using the methods described in  the manuscript is it possible to have a conclusion about the effect of amount of initial retained austenite level to the strenght and ductility of the steel ?

Yes, it is possible to qualitatively relate the amount of retained austenite to the level of mechanical properties and their variations. This investigation of structure-property relationship is the subject of another manuscript which is under preparation by authors for publication. Normally, we should expect ultra-high strength steels with enhanced ductility, particularly good uniform elongation, and improved formability and strain hardening capacity.

Reviewer 4 Report

In the introduction it should also be stated what are the different problems with using electron microscopy techniques for identification of the different phases in steels. Please check recent article who have address this issue such as Wire and arc additive manufacturing of HSLA steel: Effect of Thermal Cycles on Microstructure and Mechanical Properties.

Another critical aspect that is used in the advanced characterization of steels is atom probe tomography. The introduction should also reflect it. See for example Competition between formation of carbides and reversed austenite during tempering of a medium-manganese steel studied by thermodynamic-kinetic simulations and atom probe tomography.

The determination of the transformation temperatures in steels using the Gleeble is sensitive to the heating rate (see for example Austenite reversion kinetics and stability during tempering of a Ti-stabilized supermartensitic stainless steel: Correlative in situ synchrotron x-ray diffraction and dilatometry). This should be discussed and the heating rate used should be provided.

Why two softwares were used for phase quantification (I’m guessing Rietveld)? At least one fitted plot with the residuals should be provided.

“somewhat darker”: this is not scientific wording.

The authors mentioned several times high resolution SEM images but the images seen very pixelated, please correct

Why such a large standard deviation in all measurements of  table 3? Where is this value coming from?

No detailed about the magnetic analysis are provided in the experimental sections.

Also the mechanisms for the formation of retained austenite are not discussed in the manuscript. See for example Study of MA effect on yield strength and ductility of X80 linepipe steels weld.

Author Response

Answer to comments of reviewer 4

Comment 1:In the introduction it should also be stated what are the different problems with using electron microscopy techniques for identification of the different phases in steels. Please check recent article who have address this issue such as Wire and arc additive manufacturing of HSLA steel: Effect of Thermal Cycles on Microstructure and Mechanical Properties

Thanks for bringing to our notice the article. The obtained results indicate that the use of FE-SEM micrographs are not suitable for distinguishing finely divided retained austenite from martensite in M/RA microconstituents, even though a good contrasting distinction can be made between bainite and martensite packets. Likewise, EBSD measurements suggest that the retained austenite finer than 100 nm width can be easily missed because of the limit of resolution predicting fractions, which can be far from the actual contents, more accurately determined by the XRD.

Comment 2:Another critical aspect that is used in the advanced characterization of steels is atom probe tomography. The introduction should also reflect it. See for example Competition between formation of carbides and reversed austenite during tempering of a medium-manganese steel studied by thermodynamic-kinetic simulations and atom probe tomography.

Atom Probe Tomography (APT or 3D Atom Probe or LEAP) is the only material analysis technique offering extensive capabilities for both 3D imaging and chemical composition measurements at the atomic scale (around 0.1-0.3nm resolution in depth and 0.3-0.5nm laterally) rather than phase identification or at least estimation of its fraction. Nevertheless, it can be used for analyzing carbon and other alloying elements in retained austenite. It is also possible to detect clustering of carbon atoms or formation of carbides that may manifest during partitioning. In particular, the technique can be a useful tool to know the partitioning of Mn in medium-Mn steels and its substitution of Fe and other atoms (such as Ni and Mo) in the 3D arrangement.  Unfortunately, there was no access to this facility at the time of this research study.

Comment 3:The determination of the transformation temperatures in steels using the Gleeble is sensitive to the heating rate (see for example Austenite reversion kinetics and stability during tempering of a Ti-stabilized supermartensitic stainless steel: Correlative in situ synchrotron x-ray diffraction and dilatometry). This should be discussed and the heating rate used should be provided.

In this research work, we used 0.2oC/s on heating and 150oC/s on cooling rates for determining the critical transformation temperatures of Ac1, Ac3 and Ms, respectively. These rates were provided in section 2.1. Faster heating rates can slightly increase the Ac1 and Ac3 temperatures compared to the experimentally observed values in these tests. But the main objective of this work was to quench the samples (150°C/s) to a temperature in the bainitic range close to Ms and to understand phase transformation behaviour and the carbon enrichment of untransformed austenite as a function of time, while other competing processes such as tempering of bainite, carbide formation, fresh bainite and martensite formation during final cooling may also take place.  

Comment 4:Why two softwares were used for phase quantification (I’m guessing Rietveld)? At least one fitted plot with the residuals should be provided.

We used two software for qualitative and quantitative investigation. X-pert Highscore was used for peak identification while Maud software was used for determining the volume fraction of retained austenite and also lattice parameter for determining the carbon content of this phase by using formula in ref. 55.

Here you can find one of the fitted done.  

Comment 5: “somewhat darker”: this is not scientific wording.

 “Somewhat dark/darker” was replaced by “slightly blakish”.

Comment 6:The authors mentioned several times high resolution SEM images but the images seen very pixelated, please correct.

“High resolution SEM images” was replaced by “SEM images” in the manuscript.

Comment 7:Why such a large standard deviation in all measurements of  table 3? Where is this value coming from?

The numbers presented in Table 3 are error ranges and not standard deviations for retained austenite volume fractions measured by different techniques. These values result from the volume fraction measurements according to the following formula:

Error range= maximum{(maximum volume fraction –average volume fraction) and (average volume fraction- minimum volume fraction}

Comment 8:No detailed about the magnetic analysis are provided in the experimental sections.

The theory and details of quantitative magnetic detection technique to determine the retained austenite is quite complicated and lengthy, and is not the main topic of the present manuscript. Therefore, only a brief discription of this technique has been presented in the Introduction section.

Comment 9:Also the mechanisms for the formation of retained austenite are not discussed in the manuscript. See for example Study of MA effect on yield strength and ductility of X80 linepipe steels weld.

It was mentioned in the manuscript that “Retained austenite is stabilized by carbon redistribution and partitioning from bainite crystals to the adjacent remaining prior austenite during 350oC isothermal bainitic holding time. In fact, carbon atoms are rejected from bainite crystals into adjacent prior austenite when bainitic transformation occurs. As a result, the carbon concentration of prior austenite is increased and, consequently, its Ms temperature is decreased below room temperature. Therefore, some retained austenite is stabilized after the final water quenching step”.

Round 2

Reviewer 1 Report

The authors have made an attempt to correlate their DTA and EDX/EBSD data. In order to be fully convincing the study should really include supporting TEM data. Nevertheless it does significantly raise the interest level of the manuscript.

Author Response

In agreement with the reviewer’s comment about the TEM data supporting the measurements made through other techniques, we have already begun the process of  TEM studies of select samples,  but the analysis and interpretation of the results will take its own time. It is, therefore, not possible to provide the TEM data in the present manuscript because they are not fully completed and interepreted yet.

Reviewer 2 Report

Authors corrected the manuscript according to the recommendations.

Author Response

The authors thank the reviewer for devoting his time to review the manuscript and making recommendations.

Reviewer 4 Report

I believe that the authors have misinterpreted some of the comments raised by the reviewer. See below:

C1 – It was asked for the authors to update the literature stating the problems of using electron microscopy techniques in the identification of the steel microconstituents. Some references describing this issue were provided and the authors should update the introduction accordingly.

C2 – similar to C1 comment. It was not asked to use APT, since this is not widely available. It was requested to also bring the attention to the people who will be reading this paper about the pros and cons of such technique. Again, some key reference works have been given addressing this issue. Please update accordingly.

C3 – again, it was not asked to use different heating rates in the work, rather it was mentioned that it was important to state that the heating rate indeed influences the transformation temperatures and that this should be acknowledged. See suggested references in the first round of review and discuss properly.

C8 – this is very important to be described. If other researchers want to perform similar work as this in order to validate such finding they should be able to follow an experimental protocol. Please correct.

C9 – see the comments for C1 to C3 and update please. Or in other words, where did such reasoning come from? Did the authors discovered this for the first time, if not referencing should be made (see suggested reference in previous round).

Figure 3 is missing 3 scales. Correct it.

Author Response

Answer to comments of reviewer 4

I believe that the authors have misinterpreted some of the comments raised by the reviewer. See below:

Round 1:Comment 1: In the introduction it should also be stated what are the different problems with using electron microscopy techniques for identification of the different phases in steels. Please check recent article who have address this issue such as Wire and arc additive manufacturing of HSLA steel: Effect of Thermal Cycles on Microstructure and Mechanical Properties

Round 2:Comment 1: It was asked for the authors to update the literature stating the problems of using electron microscopy techniques in the identification of the steel microconstituents. Some references describing this issue were provided and the authors should update the introduction accordingly.

After carefully reading the suggested above-mentioned reference [49], we now realize that the reviewer has rightly pointed out the problems associated with using the electron microscopy techniques for identification of the different phases in steels as outlined in the following paragraph. Accordingly, this has been inserted in the “Introduction” section of the manuscript and the suggested article referred to. Please see the lines 86 to 99.   

“Owing to the possibilities of a number of phase constituents in steels such as ferrite, bainite, martensite and retained austenite, identification of the microconstituents by electron microscopy techniques is often problematic. As stated in [49], one of the shortcomings of EBSD in the quantification of low alloy steel microstructures, concerns the presence of different morphologies of ferrite that may have similar crystallographic structure and hence, can lead to erroneous interpretations. Bainitic microstructures also have lower confidence indexes owing to the presence of large number of dislocations, which is responsible for the high strength and low ductility typical of this phase. To some extent, the scanning electron microscopy is able to distinguish between various morphologies of ferrite and bainite in the microstructures, as depicted in Figs. 4c and 4d. However, SEM images are often not able to identify and distinguish between martensite and adjacent retained austenite,  especially in M/RA islands, as mentioned in this manuscript. But these can, however, be used to make a distinction between martensite matrices and bainite crystals in the low alloy microcomposite microstructures.”

Round 1: Comment 2: Another critical aspect that is used in the advanced characterization of steels is atom probe tomography. The introduction should also reflect it. See for example Competition between formation of carbides and reversed austenite during tempering of a medium-manganese steel studied by thermodynamic-kinetic simulations and atom probe tomography.

Round 2: Comment 2: similar to C1 comment. It was not asked to use APT, since this is not widely available. It was requested to also bring the attention to the people who will be reading this paper about the pros and cons of such technique. Again, some key reference works have been given addressing this issue. Please update accordingly.

According to the reviewer’s comment, the explanation about the APT as a powerful analyzing technique for phase or precipitate composition down to atomic scale has been added to the end of paragraph 1 in the “Introduction” section and the suggested paper referred to. Please consider lines 63 to 69.

Round 1: Comment 3: The determination of the transformation temperatures in steels using the Gleeble is sensitive to the heating rate (see for example Austenite reversion kinetics and stability during tempering of a Ti-stabilized supermartensitic stainless steel: Correlative in situ synchrotron x-ray diffraction and dilatometry). This should be discussed and the heating rate used should be provided.

Round 2: Comment 3: again, it was not asked to use different heating rates in the work, rather it was mentioned that it was important to state that the heating rate indeed influences the transformation temperatures and that this should be acknowledged. See suggested references in the first round of review and discuss properly.

The above mentioned point including the influence of heating rate on determination of critical transformation temperatures has now been stated in the “Materials and experimental procedures” section and the proposed reference has been made. Please see lines 178 to 181.

Round 2: Comment C8: No detailed about the magnetic analysis are provided in the experimental sections.

Round 2: Comment C8: this is very important to be described. If other researchers want to perform similar work as this in order to validate such finding they should be able to follow an experimental protocol. Please correct.

The following information and explanation about measuring the saturation magnetization and coercivity parameters used in the present research in order to detect the paramagnetic retained austenite in the steel samples have been added to the “Introduction” section, lines 123 to 144.

A VSM is a device that can measure the magnetic properties of a sample by its controlled movement inside a uniform applied magnetic field. The voltage induced by this movement can be detected by the pickup coils and is directly proportional to the extent of magnetization of the sample. The saturation magnetization ?? refers to the maximum magnetization in a material, which occurs when a large enough field is applied. The material only has one domain remaining in which all magnetic moments are aligned parallel to the magnetic field. Any difference in saturation magnetization between a sample containing some retained austenite and another without any retained austenite is directly related to the volume fraction of retained austenite phase. Therefore, the samples bearing some retained austenite will have higher saturation magnetization values in comparison with those without any retained austenite. In magnetization method, the saturation magnetization can be obtained from the maximum value of magnetization (Y-axis) in Hysteresis Loops, and hence, the difference in saturation magnetization value from that estimated on a fully ferromagnetic sample is directly related to the volume fraction of non-ferromagnetic retained austenite. This is due to the fact that ferrite, carbide, pearlite, bainite and martensite are ferromagnetic at low temperatures, while only the retained austenite is paramagnetic in the micro-composite microstructures [12,32–34]. Moreover, the coercivity, also called the magnetic coercivity (Hs), a measure of the ability of a ferromagnetic material to withstand an external magnetic field without becoming demagnetized is another magnetic parameter which can be used in order to detect the paramagnetic retained austenite in the samples microstructures. Experimentally, the coercivity can be measured by a horizontal intercept of the Hysteresis Loop [28,33,37].

Round 1: Comment C9: Also the mechanisms for the formation of retained austenite are not discussed in the manuscript. See for example Study of MA effect on yield strength and ductility of X80 linepipe steels weld.

Round 2: Comment C9: see the comments for C1 to C3 and update please. Or in other words, where did such reasoning come from? Did the authors discovered this for the first time, if not referencing should be made (see suggested reference in previous round).

The mechanism for the formation and/or stabilization of retained austenite and its morphology have been discussed in the manuscript in lines 386 to 394. According to the reviewr’s comment, the suggested article was referred at the end of this paragraph, line 394.

Round 2: Comment #: Figure 3 is missing 3 scales. Correct it.

Figure 3 is revised accordingly.

Round 3

Reviewer 4 Report

The manuscript was improved.